

# Evaluation of the distributions of hydroxylated isoprenoidal GDGTs in Holocene Baltic Sea sediments for reconstruction of sea surface temperature: The effect of changing salinity

Jaap S. Sinninghe Damsté[1,2], Lisa A. Warden[1], Carlo Berg[3,§], Klaus Jürgens[3], and Matthias Moros[3]

[1] Department of Marine Microbiology and Biogeochemistry, NIOZ Netherlands Institute for Sea Research, PO Box 1790 AB 59, Den Burg, The Netherlands
[2] Faculty of Geosciences, Department of Earth Sciences, Utrecht University, P.O. Box 80.021, 3508 TA Utrecht, The Netherlands
[3] Departments of Biological Oceanography and Marine Geology, Leibniz Institute for Baltic Sea Research Warnemünde (IOW), Seestraße 15, D-18119 Rostock, Germany
§ present address: Public Health Agency of Sweden, Nobels väg 18, Solna 17182 Stockholm, Sweden

*Correspondence to*: Jaap S. Sinninghe Damsté (jaap.damste@nioz.nl) or Matthias Moros (matthias.moros@io-warnemuende.de)

**Abstract.** Hydroxylated glycerol dibiphytanyl glycerol tetraethers (OH-GDGTs) produced by both marine and freshwater thaumarchaea are increasingly used for the reconstruction of past sea surface temperature (SST). They occur throughout the modern Baltic Sea, but it is unknown if their OH-GDGTs can be used for assessing past SST in this area, where salinity has changed considerably over the Holocene. Three commonly applied OH-GDGT for SST reconstruction, i.e., the OH-GDGT%, RI-OH, and RI-OH´ indices, were tested using a Thaumarchaeotal culture enriched from the Baltic Sea grown at 4 and 22°C,
and 12 surface sediments from the Baltic Sea and the adjacent Skagerrak. In the culture experiments all three proxies showed the expected response with the rise of temperature, but their absolute values were not always in line with existing marine core top calibrations, especially for the OH-GDGT% index. Of the two proxies based on the distribution of OH-GDGTs, the RI-OH index shows no increase with increasing mean annual SST, whilst the RI-OH´ index shows a linear correlation with SST when the sediments from the Bothnian Sea and Bay area, with a reduced salinity and increased lateral sediment influx, are
omitted. Two sedimentary Holocene records from the Arkona and Gotland basins were studied, the latter in high resolution. In the brackish phase of the Baltic Sea (the Littorina Sea stage), the RI-OH´ index shows a good correlation with the $TEX_{86}^{L}$, an established temperature proxy in the Baltic Sea, and can be used to identify important climatic events. However, during the preceding Ancylus Lake phase the RI-OH´ (and RI-OH) index records far too high values, resulting in anomalously high SST estimates. This is probably because freshwater thaumarchaea adjust their OH-GDGTs differently, as has been shown for
thaumarchaea in Lake Lugano's water column. In the Littorina Sea, the Ancylus Lake and the Yoldia Lake phases of the Baltic basin, the record of the RI-OH´ index, thus, most likely reflects both changes in temperature and salinity. Overall, our study indicates that a reduced salinity increases the values of the RI-OH´ (and RI-OH) indices substantially and this should be considered when applying these proxies in other settings.



# 1 Introduction

Thaumarchaea are archaea that occur ubiquitously in aquatic environments and are mostly acting as nitrifiers (see Offre et al., 2013 for a review) and thus play an important role in the biogeochemical nitrogen cycle. They use glycerol dibiphytanyl glycerol tetraethers (GDGTs) as their membrane-spanning lipids, which can contain a varying number of cyclopentane moieties (see, Schouten et al., 2013 for a review). In addition, they contain crenarchaeol, a structurally unique GDGT with one cyclohexane and four cyclopentane moieties (Sinninghe Damsté et al., 2002; Holzheimer et al., 2021). Like most archaea (De

Rosa and Gambacorta, 1988), thaumarchaea adjust the number of cyclopentane moieties depending on growth temperature (Wuchter et al., 2004). The increase in the production of cyclopentane rings with higher temperatures is thought to be a mechanism of temperature adaptation within the cell membrane, for example to maintain membrane permeability and fluidity at various temperatures (Gabriel and Chong, 2000; Gliozzi et al., 1983). Because of this physical adaptation, the distribution of thaumarchaeotal GDGTs in marine (Schouten et al., 2002) and lacustrine (Powers et al., 2004) sediments can be used to

estimate past water temperature using the $TEX_{86}$ (TetraEther indeX), which is based on the relative abundance of specific GDGTs, and shows a relationship to sea surface temperature (SST) (Kim et al., 2008). The relationship between the number of cyclopentane moieties, expressed by the $TEX_{86}$, and SST is weaker in cooler waters (i.e., below 15 °C,) and a modified index, termed $\mathbf{TEX_{86}^{L}}$, was developed to reconstruct SST for the lower temperature range (Kim et al., 2010). However, reconstruction of SST in the polar oceans using these indices is often faced with difficulties (see, Fietz et al., 2020, for a

review).

Liu et al. (2012) identified a new type of GDGT, i.e., hydroxylated GDGTs (OH-GDGTs), in marine sediments from both tropical and temperate regions. These GDGTs contain 0-2 cyclopentane moieties with the hydroxyl group positioned at one of the tertiary carbon atoms of one of the two biphytanyl chains that is closest to the glycerol moiety of the GDGT (for structures see, Liu et al., 2012). It was postulated that their distribution could potentially be used as a proxy for SST, which was supported

through the exploration of the distribution of OH-GDGTs in marine subsurface sediments (Liu et al., 2012). The addition of a hydroxyl group to GDGTs is thought to be an adaptation to the cold to adjust the permeability of the membrane. Hence, inclusion of OH-GDGTs in GDGT-based palaeotemperature proxies may aid in SST reconstruction in polar oceans (Fietz et al., 2020). Huguet et al. (2013) showed that OH-GDGTs occur widely in both marine and lacustrine sediments and that the abundance of OH-GDGTs relative to non-hydroxylated GDGTs (expressed by the so-called OH-GDGT% index) correlated

with increasing latitude and lower SST, suggesting the potential to serve as a novel paleotemperature proxy. Fietz et al. (2013) subsequently proposed two other indices based on OH-GDGTs, which were used to reconstruct SST in Nordic Seas as an alternative for $\mathbf{TEX_{86}^{L}}$ and $TEX_{86}$, and subsequently in the Arctic Ocean (Knies et al., 2014). A study of an extended set of surface sediments from the Chinese coastal seas demonstrated that the distribution of the three OH-GDGTs themselves is also related to SST, which led to the definition of two ring indices of hydroxylated tetraethers (RI-OH and RI-OH') (Lü et al., 2015).

These indices have subsequently been successfully applied to reconstruct past SST in open marine systems, such as the Gulf



of Lion (Davtian et al., 2019), the Iberian Margin (Davtian et al., 2021), and the Alboran Sea (Morcillo-Montalbo et al., 2021), allowing to perform SST reconstruction based on several different organic paleothermometers.

A potential bias in the use of OH-GDGTs in palaeothermometry in marine sediments may be the input of OH-GDGTs from soil and freshwater systems. Kang et al. (2017) showed that OH-GDGT-2 (i.e., the OH-GDGT with two cyclopentane moieties) was relatively more abundant in soils and lake sediments than in the suspended matter of the estuary of the Han River and the sediments of the adjoining Yellow Sea. Lü et al. (2019) noted lower concentrations of OH-GDGTs in suspended particulate matter (SPM) of the Yangtze River estuary in winter than in summer and in the lower reaches of the river than in the estuary, suggesting a predominant source by in situ production of marine thaumarchaeota in summer. However, in the stratified, anoxic Lake Lugano a single thaumarchaeote thriving in the suboxic waters below the thermocline (30-70 m) also produced OH-GDGTs that were transported to the lake floor as revealed by sediment trap studies (Sinninghe Damsté et al., 2022), indicating that OH-GDGTs may also be formed in freshwater environments.

Due to the variable salinity of the Baltic Sea during the Holocene, complications arise when attempting to use the alkenone unsaturation indices to reconstruct SST (Blanz et al., 2005; Schulz et al., 2000; Warden et al., 2016). Previous studies have successfully reconstructed summer SST in the Baltic Sea using a calibration specific to the region based on the $\mathbf{TEX^L_{86}}$ index during the last 7.2 cal kyr, when the Baltic Sea was connected to the open ocean and its waters were brackish (Kabel et al., 2012; Warden et al., 2017), but not for the preceding freshwater phase. OH-GDGTs may potentially be used as alternative paleotemperature proxies in this area. Kaiser and Arz (2016) studied a large set (n=57) of surface sediments from the entire Baltic Sea and the adjoining Skagerrak area for OH-GDGTs and GDGTs. The concentration of the OH-GDGTs varied from 6-140 µg g$^{-1}$ TOC (average ca. 40 µg g$^{-1}$ TOC), representing 8-14% of the total GDGTs and OH-GDGTs. OH-GDGT-0 was the most dominant OH-GDGT, typically representing ca. 85% of all OH-GDGTs. High positive linear correlations ($R^2 = 0.97$-0.99) were observed between the concentrations of crenarchaeol and the three OH-GDGTs, which strongly suggested a common autochthonous source, likely Thaumarchaeota. Indeed, Thaumarchaeota of the marine group 1.1a, the *Nitrosopumilus* lineage, are abundant in the central Baltic Sea basins above the redoxcline, contributing 20-30% of total prokaryotes (Labrenz et al., 2010), and being responsible for nearly all the ammonia oxidation in these water layers (Berg et al., 2015b). One representative of this lineage, closely related to *Ca.* Nitrosoarchaeum limnia (Blainey et al., 2011), has been enriched, and examined in laboratory experiments (Berg et al., 2015a), and forms a likely source for the (OH)-GDGTs in Baltic Sea sediments. A combined lipidomic and 16S rRNA gene amplicon sequencing approach in the suboxic waters (80–110 mwd) of the Black Sea, in many aspects a similar environment as the stratified anoxic basins of the Baltic, revealed that marine group 1.1a thaumarchaea dominated and produced predominantly GDGTs and OH-GDGTs with monohexose polar head groups (Sollai et al., 2019).

Here we study the presence of OH-GDGTs in a thaumarchaeal enrichment culture from the Baltic Sea (Berg et al., 2015a) and determine their presence and distribution in its surface sediments and the potential of the OH-GDGT%, RI-OH, and RI-OH'



proxies for SST estimation. Our major goal was to evaluate Holocene sedimentary records of OH-GDGTs in the Gotland and Arkona Basins for potential application as SST proxies.

## 2 Setting

The present-day Baltic Sea (Fig. 1) is almost completely landlocked except for a narrow connection to the North Sea through the Straits of Denmark (Kattegat and Skagerrak), where inflows of denser saline bottom waters can occur. These influxes of marine waters cause a lateral salinity gradient that decreases from the south to the north Baltic Sea, where there is a large input of fresh water from rivers. The Baltic Sea has gone through several phases since the last deglaciation. When the large Scandinavian ice caps retreated, a freshwater lake known as the Baltic Ice Lake evolved in the Baltic Sea basin. This lake was disconnected from the North Sea. The subsequent further advance of the ice sheet retreat established a connection with the North Sea, causing an influx of marine water and creating the slightly brackish conditions of the Yoldia Sea (Björck, 1995; Jensen, 1995). As the ice sheet melted, the landmass was lifted (i.e., the so-called isostatic rebound) causing the basin again to be disconnected from the North Sea (Jensen et al., 1999). This separation from ocean waters produced an enclosed freshwater basin, the Ancylus Lake (Björck, 1995). Subsequent eustatic sea-level rise caused the connection to the North Sea to become re-established and allowed once again for the inflow of saline water transforming the lake into the Littorina Sea (Winterhalter, 1992). The transformation of the Ancylus Lake to the Littorina Sea is considered to be very complex consisting of pulses of marine waters entering the basin for nearly 2,000 years before becoming a fully brackish basin (Andren et al., 2000).

## 3 Material and Methods

### 3.1 Enrichment culture

A Thaumarchaeotal enrichment culture, closely related to *Ca.* Nitrosoarchaeum limnia obtained from the Landsort Deep in the Baltic Sea (Berg et al., 2015a), was grown in filtered seawater from the same location at different temperatures (4 and 22 ºC) in the dark but at constant salinity (12 g kg$^{-1}$) to determine the effect of growth temperature on the abundance relative to the GDGTs and the distribution of OH-GDGTs. After reaching stationary phase, the cultures were filtered onto 0.22 μm filters and stored at -20 °C until analysis.

### 3.2 Sampling of sediments

Ten surface (0-0.5 or 0-1 cm; Table 1) sediments from different sub-basins of the Baltic Sea and one from the Skagerrak, a sea strait that connects the Baltic Sea with the North Sea (Fig. 1), were sampled from multi-cores obtained during various cruises by carefully cutting the cores from top to bottom in 0.5 or 1 cm slices. An additional surface sediment was collected using a multi corer during cruise EMB046 of the R/V *Elisabeth Mann-Borgese* in May 2013.



Five sediment cores from two locations were used in this study (Table 1; Fig. 1). A 52 cm-long core (P435-1-4MUC) was collected using a multi corer from the Gotland Basin in the central Baltic Sea during the R/V "Poseidon" cruise P435 in June 2012. The sediment core, which contained the water-sediment interface, was sub-sampled and analyzed at 0.5 cm resolution. Two longer piston cores, 303600-N and 303600-3, were obtained from the same location during the cruise campaign on the

R/V *Prof. Albrecht Penck* in July 2007. The 377 cm-long core 303600-N was cut into 1-cm sections from 0-237 cm and the remainder of the core was cut into 2-cm sections. In total, 300 samples (excluding duplicates) were analyzed. The core was correlated to P435-1-4 MUC as described previously (Warden et al., 2018). Core 303600-3 was ca. 820 cm long and was correlated to core 303600-N by comparison of loss on ignition and XRF core scanner profiles. A selected number (*n*=31) of 1 cm slices from core 303600-3 were analyzed. Data obtained from the analysis of these three cores were combined to form a

Gotland Basin composite record, which spans a part of the Yoldia phase, the Ancylus Lake phase (10.6-7.8 cal kyr BP), the Ancylus Lake/Littorina Sea transition (7.7-7.3 cal kyr BP), and the recent brackish phase (7.2 cal kyr BP-2008 AD) (see for details, Sollai et al., 2017; Warden et al., 2018). Sediment cores 318310 (0–1160 cm) and 318340 (0–670 cm) were collected using a gravity corer on the R/V *Maria S. Merian* from the Arkona Basin in April 2006 (Table 1; Fig. 1). The correlation methods and core depth age models for these cores have been described in Warden et al. (2016). Data obtained from 1 cm-

sediment slices (*n*=44) from the two cores were combined to form a composite sedimentary record including the Baltic Ice Lake phase, the Yoldia Sea phase, Ancylus Lake phase, the Ancylus Lake/Littorina Sea transition and the recent brackish phase.

**3.3 Lipid extraction and GDGT analysis**

The filters of the cultures were extracted with a modified Bligh and Dyer extraction and the extracts were base hydrolysed to

release core GDGTs as previously described (Pitcher et al., 2011).

All sediment samples were freeze-dried and then subsequently grounded and homogenized by means of a mortar and pestle. These samples were then extracted with the Dionex$^{TM}$ accelerated solvent extraction (ASE) using 1-3 g of sediment and extracting with dichloromethane (DCM):methanol (9:1, v/v) as the solvents at a temperature of 100 °C and a pressure of 1,500 psi for 5 min with 60% flush and purge 60 s. The extract was collected and then dried using Caliper Turbovap®LV. Next,

using DCM, the lipid extract was dried over a column of anhydrous $Na_2SO_4$ and then blown down under a gentle stream of $N_2$. To quantify the amount of GDGTs, 1 µg of an internal standard (a $C_{46}$ glycerol trialkyl glycerol tetraether; Huguet et al., 2006) was added to the total lipid extract, which was subsequently separated over an $Al_2O_3$ column (activated for 2 h at 150 °C) into three fractions using hexane:DCM (9:1, v:v), hexane:DCM (1:1, v:v), and DCM:MeOH (1:1, v:v) to obtain the apolar, ketone and polar fractions, respectively.

The polar fractions of the extracts of both the cultures and the sediments, which contain the GDGTs, were dried using $N_2$, re-dissolved in hexane:isopropoanol (99:1, v:v) at a concentration of 10 mg ml$^{-1}$ and then passed through a 0.45 µm PTFE filter and analyzed with an Agilent 1290 Infinity ultra-high performance liquid chromatography (UHPLC) coupled to an Agilent 6130 single quadrupole mass detector according to Hopmans et al. (2016) with single ion monitoring of [M+H]$^+$ ± 0.5 *m/z*




units. GDGTs were quantified by integration of the *m/z* 1302, 1300, 1298, 1296, and 1292 mass chromatograms. The later
eluting OH-GDGTs were quantified by integration of the [M+ H]$^+$ - 18 fragments, observed in the *m/z* 1300, 1298, and 1296
mass chromatograms, as was done in previous studies (Huguet et al., 2013; Lü et al., 2015; Kaiser and Arz, 2016).

**3.4 Calculation of GDGT-based indices and SST-GDGT proxy relationships**

The percentage of OH-GDGTs compared with the total GDGTs was calculated according to Huguet et al. (2013):

OH-GDGT% = Σ[OH-GDGTs] / {Σ[OH-GDGTs] + Σ[GDGTs]} × 100  (1)

Where the Σ[OH-GDGTs] comprises the summed abundances of OH-GDGT-0, OH-GDGT-1 and OH-GDGT-2, and
Σ[GDGTs] reflects the summed abundances of GDGT-0, GDGT-1, GDGT-2, GDGT-3, crenarchaeol and its isomer.

The RI-OH and RI-OH' indices used for SST determination were calculated according to Lü et al. (2015):

RI-OH = {[OH-GDGT-1] + 2*[OH-GDGT-1]} / {[OH-GDGT-1] + [OH-GDGT-2]}  (2)

RI-OH' = {[OH-GDGT-1] + 2*[OH-GDGT-1]} / Σ[OH-GDGTs]  (3)

The TEX$_{86}$ was calculated according to Schouten et al. (2002):

TEX$_{86}$ = {[GDGT-2]+[GDGT-3]+[Cren']}/ {[GDGT-1]+[GDGT-2]+[GDGT-3]+[Cren']}  (4)

The TEX$_{86}^L$ was calculated according to Kim et al. (2010):

TEX$_{86}^L$ = [GDGT-2]/ {[GDGT-1]+[GDGT-2]+[GDGT-3]}  (5)

Analysis of duplicates (n=15) revealed that the differences for OH-GDGT%, RI-OH and RI-OH' were on average, 2.2, 0.3 and
1.4 %, respectively.

RI-OH and RI-OH' values were converted with the relationship proposed by Lu et al. (2015) and Fietz et al. (2020),
respectively:

RI-OH' = 0.0422 * SST – 0.029  (6)

RI-OH = 0.018 * SST + 1.11          (R$^2$ = 0.74, *n* = 107)  (7)

where SST is annual mean sea surface temperature.

TEX$_{86}^L$ values were converted to July to October SSTs with the Baltic Sea calibration proposed by Kabel et al. (2012):

T = 34.03* TEX$_{86}^L$ + 36.73 (R$^2$ = 0.89, *n* = 9)  (8)

**4 Results**

**4.1 Occurrence and distribution of OH-GDGTs in a thaumarchaeotal enrichment culture**

A thaumarchaeotal culture enriched from Baltic Sea water and closely related to *Ca.* Nitrosoarchaeum limnia (Berg et al.,
2015a) was grown at 4 and 22 °C but otherwise identical conditions. The distribution of the core GDGTs was studied after
base hydrolysis of the Bligh-Dyer extract to release the polar head groups of the GDGTs. The hydrolysate was dominated by
GDGT-0 and crenarchaeol (Table 2) as is commonly observed for thaumarchaea (see, Bale et al., 2019 for a recent overview).





A marked difference was noted between the different growth temperatures; at 22 °C cyclopentane-containing GDGTs were

relatively more abundant than at 4 °C. This resulted in an increase of the $TEX_{86}$ from 0.32 at 4°C to 0.49 at 22°C. OH-GDGTs were detected in relatively low amounts (OH-GDGT% = 1.3-2.4; Table 2) with higher relative amounts at the low growth temperature. A marked change in the distribution of the OH-GDGTs was also apparent. At 4 °C, OH-GDGT-0 was most abundant, whereas OH-GDGT-2 dominated at 22°C (Table 2). This change in distribution resulted in increasing RI-OH and RI-OH' indices, i.e., from 1.19 to 1.74, and 0.23 to 1.50, respectively (Table 2).

### 4.2 Occurrence and distribution of OH-GDGTs in surface sediments

All three known OH-GDGTs were detected in the Baltic Sea and Skagerrak surface sediments (Fig. 2). OH-GDGT% ranged between 4.7-8.6 (Table 1) and for the Baltic Sea was on average $7.2 \pm 0.7$ SD (standard deviation) with no clear trend. This was lower than the $10.0 \pm 1.0$ SD reported by Kaiser and Arz (2016) for surface sediments from the Baltic Sea. The surface sediments of the stations in the Gulf of Finland (349190, 341200; Fig. 1) and that of the most southern station (377850; Fig.

1) had the highest OH-GDGT% (8.3-8.6). OH-GDGT-0 made up the largest component of the OH-GDGTs with an average fractional abundance of $0.85\pm0.04$ SD. OH-GDGT-1 was the second most abundant at $0.12\pm0.03$ SD and OH-GDGT-2 had the lowest fractional abundance at $0.03\pm 0.01$ SD. This is comparable to previously reported (Kaiser and Arz, 2016) fractional abundances: OH-GDGT-0 $0.84\pm0.02$ SD, OH-GDGT-1 $0.13\pm0.02$ SD, and OH-GDGT-2 $0.04\pm0.01$. The average RI-OH and RI-OH' indices for the Baltic Sea surface sediments was $1.22\pm0.04$ SD, and $0.18\pm0.05$ SD, respectively, which is slightly

higher and lower, respectively, as for the two sediments from the Skagerrak. The average $TEX_{86}$ values for the Baltic Sea sediments was $0.41\pm0.03$ SD, slightly higher than that for the two Skagerrak sediments (Table 1).

### 4.3 Occurrence and distribution of OH-GDGTs in the Holocene sedimentary record

Three sediment cores from the Gotland Basin from two nearby stations (Table 1) were examined for OH-GDGTs. The sections covered by the three cores partially overlap, allowing to obtain one combined sedimentary record covering part of the Yoldia

Sea phase, the freshwater period of the Baltic Sea, the Ancylus Lake phase, and the brackish Littorina Sea phase. The TOC profiles (Fig. 2a) reported previously (Warden et al., 2017, 2018) reveals peak values at the modern warm period (MoWP), the medieval climate anomaly (MCA), and the Holocene thermal maximum (HTM); these periods of TOC enrichment co-vary with the occurrence of strong lamination, both caused by the occurrence of bottom water anoxia. The Yoldia Sea and Ancylus Lake phases are characterized by much lower TOC values. TOC values are not available for the lower part of the section but

were estimated from the loss-on-ignition (LOI) profile and reveal a continuously lower TOC content in lower part of the section (Fig. 2a).

The concentration of the summed OH-GDGTs varied from 5 - ~100 µg g$^{-1}$ TOC (Fig. 2c). There was one outlier at 182 cm, where the concentration was 201 µg g$^{-1}$ TOC. Overall, the concentration profile was similar to that of crenarchaeol (Fig. 2b), which was previously reported by Warden et al (2018). Indeed, the concentrations of crenarchaeol and summed OH-GDGTs

showed a positive, linear correlation ($R^2 = 0.95$, $n=179$). The OH-GDGT% record revealed considerable variation (Fig. 2d).

 

In the Yoldia Sea and Ancylus Lake phase, OH-GDGT% generally fluctuates between 6-10, but at the end of the Ancylus Lake phase dropped to lower values of ~5, which were also observed for the Littorina Sea phase, with only slightly higher values in the uppermost 10 cm, where OH-GDGT% exhibited similar values as the average of the surface sediments of the Baltic Sea (7.2). The sediment at 182 cm acted again as an outlier with a much higher OH-GDGT% (16.0±0.3%, triplicate analysis). In this section (140-200 cm) three other horizons also show high values for OH-GDGT% (16-25%; Fig. 2d). In these instances the concentration of the summed OH-GDGTs was not measured, and it could not be established of the high OH-GDGT% was also due to an anonymously high concentration of OH-GDGTs.

OH-GDGT-0 was the most abundant OH-GDGT; its fractional abundance varied from 0.47 – 0.89. OH-GDGT-1 was the second most abundant OH-GDGT; its fractional abundance varied from 0.09 – 0.30. OH-GDGT-2 was always the least abundant component with the lowest fractional abundance (0.02) in the brackish phase, and some parts of the Yoldia Sea phase, and highest fractional abundances (up to 0.24) in the Ancylus Lake phase. These changes in the distribution of the OH-GDGTs resulted in marked changes in the RI-OH and RI-OH´ indices. The record of the RI-OH (Fig. 2e) showed values around 1.2 for the lowermost section and the brackish section, values close to the average of the Baltic Sea surface sediments studied. At the end of the Yoldia Sea stage RI-OH increased consistently and finally reached a maximum of ~1.44 at the end of the Ancylus Lake stage (Fig. 2e). During the transition from the Ancylus Lake to the Littorina Sea stage RI-OH consistently declines to values of ~1.20. A similar trend is noted for the RI-OH´ record (*cf.*, Figs. 2e-f). The Littorina Sea phase shows relatively constant values of ~0.18, a similar value as the average of the studied surface sediments of the Baltic Sea, although the high-resolution analysis revealed small but significant variations. The Yoldia Sea stage showed slightly higher RI-OH´ values than the brackish phase but at 407-409 cm there is a marked decline to values of 0.14, which is evident in the two different cores covering this section (Fig. 2f). From this point, there was a sudden marked increase in RI-OH´ to ~0.4, followed by a more gradual increase to ~0.7 in the first part of the Ancylus Lake phase. In the first phase of the transition from the Ancylus Lake to the Littorina Sea stage rapidly declines towards RI-OH´ values of ~0.26, and, after, a small increase, fall to values of ~0.18.

Two sediment cores from the Arkona Basin from two nearby stations (Table 1) were examined for OH-GDGTs, albeit with a much coarser resolution than those of the Gotland Basin. The sections covered by these cores partially overlap and have been correlated to each other (Warden et al., 2016). This allowed to obtain one combined sedimentary record covering part of the Yoldia Sea phase, the freshwater period of the Baltic Sea, i.e., the Ancylus Lake phase, and the brackish Littorina Sea phase. Only LOI data from the upper 6 m of the section is available (Warden et al, 2016) and shows higher LOI values in the brackish phase with the highest values during the MCA.

The concentration of the summed OH-GDGTs varied from 2 - ~25 µg g⁻¹ TOC (Fig. 3c). There was one outlier at 540 cm, where the concentration was 92 µg g⁻¹ TOC. Overall, the concentration profile was like that of crenarchaeol (Fig. 3c), previously reported by Warden et al. (2018). The concentrations of crenarchaeol (Fig. 3b) and summed OH-GDGTs showed a positive, linear correlation (R² = 0.91, *n*=21). The OH-GDGT% record showed quite some variation, i.e., it ranged from 1.5 – 9.5, excluding the outlier with the extraordinary high OH-GDGT concentration. During the Baltic Ice Lake phase (3.8±0.6

SD) and the recent brackish phase (4.8±0.5 SD), the OH-GDGT% was less variable than during the Yoldia Sea phase (5.5±1.4

SD) and the Ancylus Lake phase (4.8±1.5 SD). As observed for the Gotland Basin, OH-GDGT-0 was the most abundant OH-

GDGT, except for the 700-705 cm section, where OH-GDGT-2 becomes most abundant. Its fractional abundance varies

markedly, i.e., from 0.36-0.79. OH-GDGT-1 was in most sediments the second most abundant component with fractional

abundances ranging from 0.15-0.29. The fractional abundance of OH-GDGT-2 ranged from 0.05 to 0.35, and in parts (i.e.,

520-584 cm, 614-634 cm, and 700-705 cm) of the Ancylus Lake phase, it was higher than that of OH-GDGT-1. These changes

in fractional abundance resulted in marked parallel ($R^2$ = 0.94, $n$=44) changes in the records of the RI-OH and RI-OH′ indices

(Figs. 3e-f), which are quite comparable to those of the Gotland Basin (cf., Figs. 2e-f). In the brackish phase RI-OH values are

around 1.28 and similar, although slightly more variable, values were recorded in the Ice Lake phase. An increase is noted for

the Yoldia Sea phase, culminating in substantially higher values (1.4-1.6) in the Ancylus Lake phase (Fig. 3e), which is higher

than those recorded in the Gotland Basin (Fig. 2e). The RI-OH′ record shows a similar phenomenon: relatively low but variable

values (0.29-0.44) during the Ice Lake phase, followed by an increase in the Yoldia Sea phase, subsequently reaching values

slightly >1 in the Ancylus Lake phase. These values are again higher than those recorded in the Ancylus Lake phase of the

Gotland Basin. During the transition from the Ancylus Lake to the Littorina Sea phase RI-OH′ values rapidly drop to values

of ~0.3, slightly higher than recorded in the Gotland Basin during this period.

## 5 Discussion

### 5.1 Biological sources of OH-GDGTs in the Baltic Sea

Kaiser and Arz (2016) observed high positive correlations between the concentrations of crenarchaeol and the OH-GDGTs in

surface sediments from the Baltic Sea. This strongly suggested a common thaumarchaeal source since crenarchaeol is a highly

specific membrane lipid of thaumarchaea (e.g., Sinninghe Damsté et al, 2002; Bale et al., 2019). Indeed, OH-GDGTs have

been detected in cultures of most species of the *Nitrosopumilales* (e.g., Liu et al., 2012; Elling et al., 2017; Bale et al., 2019).

Our detection of OH-GDGTs in an enrichment culture phylogenetically related to *Ca.* Nitrosoarchaeum limnia obtained from

water of the Landsort Deep in the Baltic Sea (Berg et al., 2015a) further confirms the biological source of OH-GDGTs in the

Baltic Sea (Table 1). At a growth temperature of 4 °C, the distribution of the OH-GDGTs was like that of Baltic Sea surface

sediments (Kaiser and Arz, 2016; this study), i.e., dominated by the OH-GDGT without cyclopentane moieties (OH-GDGT-

0). Somewhat surprising, however, the abundance of the OH-GDGTs relative to the other GDGTs in the enrichment culture

was lower than expected; the OH-GDGT% value of 2.4 was substantially lower than the average value measured in the surface

sediments of the Baltic Sea (7.2±0.9 SD; $n$=10) reported here (Table 1) and that (10.0±1.2 SD; $n$=42) reported previously by

Kaiser and Arz (2016). It is thought that the relative abundance of OH-GDGTs increases with decreasing temperature (Huguet

et al., 2013), so this discrepancy is hard to explain, especially since the average temperature of the Baltic Sea is higher than

the temperature (i.e., 4 °C) at which the enrichment culture was grown.





Further circumstantial evidence for a thaumarchaeal origin of the OH-GDGTs comes from the sedimentary records (Figs. 2 and 3); in both the Gotland Basin and the Arkona Basin a high positive (i.e., $R^2 > 0.94$) correlation between the concentrations of crenarchaeol and the OH-GDGTs is observed. This strongly suggests that not only in the present-day Baltic Sea, but also during the Holocene, OH-GDGTs were primarily derived from pelagic thaumarchaea.

## 5.2 Influence of temperature on the relative abundance and distribution of OH-GDGTs

### 5.2.1 Enrichment culture

An earlier study of SPM and sediments showed that with decreasing SST, OH-GDGT% increases, albeit that the relationship showed substantial scatter ($R^2 = 0.42$, $n$=72; Huguet et al., 2013). The same trend was observed in the experiments with the enrichment cultures grown at; OH-GDGT% increased from 1.3 at 22 °C to 2.4 at 4 °C, but the absolute value at 4 °C is much lower than OH-GDGT% values measured in surface sediments of cold oceans and brackish basins (Huguet et al., 2013; Kaiser and Arz, 2016). This suggests that the OH-GDGT% ratio is a problematic SST proxy, as is also evident from the scattered relationship between SST and OH-GDGT% (Huguet et al., 2013). However, the change in the distribution of OH-GDGTs with growth temperature, from OH-GDGT-0 representing the major OH-GDGT in the enrichment culture grown at 4 °C to the predominance of OH-GDGT-2 in the culture grown at 22 °C (Table 2), gives credence to the use of the distribution of OH-GDGTs as a proxy for SST (Lü et al., 2015). This change is also reflected by a change in the OH-GDGTs proxy ratios, i.e., from 1.19 to 1.74 for the RI-OH and from 0.23 to 1.50 for the RI-OH′. These values would translate with the most recent SST calibrations for RI-OH (Lü et al., 2015) and RI-OH′ (Fietz et al., 2020) into temperatures of 6 and 36 °C, and 4 and 35 °C, respectively. The high temperatures exceed the growth temperature of the thaumarchaeal culture by at least 13 °C, but it should be born in mind that the calibration for RI-OH′ was originally designed for SSTs below 15 °C (Lü et al., 2015).

The increase in cyclopentane-containing GDGTs as observed in the enrichment culture grown at the higher temperature (22 °C; Table 2) is in line with what has been previously demonstrated in marine sediments (Schouten et al., 2002). For the $TEX_{86}^L$, the GDGT-based paleothermometer that works best for the Baltic Sea (Kabel et al., 2012), values increase from -0.81 to -0.37, which translates to SSTs of 9 and 24°C, respectively. This is slightly higher than the growth temperatures, but it should be noted that Kabel et al. (2012) calibrated $TEX_{86}^L$ against the average SST over the July-October period, i.e., an SST higher than that of annual mean SST. Hence, the results of the growth experiments confirm the applicability of the $TEX_{86}^L$-SST relationship in the Baltic Sea.

### 5.2.2 Surface sediments

Previously it has been observed in marine sediments that the OH-GDGT% ratio increases with decreasing SST (Huguet et al., 2013). A high linear correlation ($R^2$=0.98) between SST and OH-GDGT% was noted for the surface sediments studied (i.e., including two sediments from the Skagerrak), but only when the three datapoints from Bothnian Sea and Bay were omitted; despite the lower SSTs of these stations in the northern Baltic Sea (Table 1), the OH-GDGT% ratio did not increase (Fig. 4a).



The Bothnian Sea and Bay are characterized by a lower salinity (3.4-5.8 g kg$^{-1}$; Table 1) than the other stations. Furthermore, an increased lateral sediment influx of old re-suspended Littorina Sea stage sediments was shown in this region (e.g., Häusler et al., 2017; Moros et al., 2020), which suggests that the OH-GDGT signal in the surface sediments may not be fully
autochthonous. In an earlier study, Kaiser and Arz (2016) did not observe lower values for OH-GDGT% in this northern part of the Baltic Sea and generally show higher values for the OH-GDGT% than reported in this study (Fig. 4a). This difference may relate to variations in the ionization efficiency between GDGTs and OH-GDGTs using different mass spectrometers. Even in our data set, where all samples were analyzed with the same instrument, we observe that the analytical error for the OH-GDGT% ratio is substantially larger than those for the RI-OH and RI-OH´ ratios, which are based on the distribution of
the OH-GDGTs (see experimental). Another concern is the large variation in this ratio in the first centimetres of the MUC core from the Gotland basin, where OH-GDGT% varied from 5.2-6.9 for unknown reasons. Hence the thickness of the sample analyzed as "surface sediment" and potential erosion of the top sediment may also substantially affect the obtained value. When the Baltic Sea OH-GDGT% data are plotted together with two global datasets (Fig. 4a), they fit the general trend of higher OH-GDGT% values with decreasing SST, but it is also clear from the large scatter that the OH-GDGT% proxy cannot
be reliably applied as a proxy to predict SST.

When we focus on the two OH-GDGT temperature proxy ratios that are based upon their distribution (i.e., RI-OH and RI-OH´), it is evident that for our set of Baltic Sea surface sediments the RI-OH ratio was constant at 1.21±0.01 SD (n=9) and showed no variation over the SST range of 5.8-9.7°C, except for one outlier at 1.34 for the central Baltic Sea (Fig. 4b; Table 1). This is comparable to the data of Kaiser and Arz (2016), where the average RI-OH ratio was 1.22±0.03 SD (*n*=41) and also
did not show a variation with SST (Fig. 4b). Hence, the RI-OH ratio does not reflect SST changes in the Baltic Sea. This is not unexpected, since Lü et al. (2015) introduced an alternative index, the RI-OH´, for cold (<15 °C) environments because the RI-OH ratio did not perform well in such settings. The RI-OH´ ratio does show a linear relationship with SST but only when the three datapoints from Bothnian Sea and Bay were omitted (Fig. 4d), like for the SST – OH-GDGT% correlation:

RI-OH´ = 0.0647 * SST - 0.4046 (R$^2$ = 0.69; *n* = 9)                                           (9)

The same observation can be made with the dataset of Kaiser and Arz (2016): surface sediments from the Bothnian Sea and Bay with SSTs ranging from 5.9-7.0 showed a random variation with the RI-OH´ ratio that varied from 0.146-0.243 (i.e., almost the same range as for all datapoints; Fig. 4c), while for the remaining datapoints there was a weak positive correlation between SST and RI-OH´. Overall, the new Baltic surface sediment data for the RI-OH´ index fits with the existing global dataset that was used to correlate SST and RI-OH´ (Fietz et al., 2020).

**5.3 Application of OH-GDGT-derived SST proxies to the Baltic Sea sedimentary records**

Next, the OH-GDGT proxies were applied to the sedimentary records of both the Gotland Basin and the Arkona Basin (Figs. 2d-f and 3d-f). The values of the OH-GDGT% index during the brackish phase in both basins were lower than the average values for the surface sediments (Figs. 2d and 3d). In the high-resolution record of the P435-1-4 MUC core of the Gotland Basin a decline is observed in the first 10 cm, which may be related to preferential destruction over the other GDGTs (i.e.,





diagenetic alteration). Indeed, the concentration of the OH-GDGTs decreases more rapidly that of crenarchaeol (cf., Figs. 2b
       and 2c). The OH-GDGT% index increased further downcore for both the Gotland and the Arkona basin records into the
       Ancylus Lake phase and fluctuates a lot compared to the records of the RI-OH and RI-OH´ indices (Figs. 2e-f), which are
       discussed below. In view of the calibration issues (Fig. 4a), the potential influence of diagenesis, the potential analytical
       problems in determining the OH-GDGT% index, and the apparent noisiness of the obtained record (Fig. 2f), we conclude that
the OH-GDGT% index is less suitable for the determination of SST.

       The records of the RI-OH and RI-OH´ indices reveal similar downcore trends in both the Gotland (Figs. 2e-f) and Arkona
       Basin (Figs. 3e-f). The clearest trend is seen in the RI-OH´ records, which will be discussed in detail. In the brackish Littorina
       phase, the RI-OH´ index shows similar values as those measured for the surface sediments, whereas they are substantially
       higher during the Ancylus Lake phase (Figs. 2f and 3f). At the transition from the Ancylus Lake phase towards the Littorina
phase, a marked change is apparent: RI-OH´ index values gradually decline from ca. 0.7 to ca. 0.2 in the interval of 250-210
       cm of the Gotland Basin record (Fig. 2f). In the Arkona Basin record the RI-OH´ values in the Ancylus Lake phase are even
       higher (0.8-1.0) and decrease to values of ca. 0.3. This decline also seems to be gradual, but the coarser sampling resolution
       does not allow to conclude this with certainty. The gradual decline of the RI-OH´ index in the 250-210 cm section of the
       Gotland Basin record matches quite well with the gradual changes observed for other biomarker (i.e., the BIT index, the
MBT´$_{5Me}$ index (Warden et al., 2018), the distribution of the cyanobacterial glycolipids (Sollai et al., 2017)) and bulk
       parameters (i.e., the TOC content and del$^{15}$N; Sollai et al., 2017) in this record. These have all been interpreted to reflect the
       change in conditions from a freshwater lake to a brackish enclosed basin established by the (re)connection with the North Sea.
       The large change in the RI-OH´ index values is rather substantial when considered from the viewpoint of paleothermometry:
       a value of 0.8 would translate into an annual mean SST of 20°C using the recent marine calibration of Fietz et al. (2020) (Fig.
4c). Even with the local Baltic Sea calibration, we arrive at an SST estimate of ca. 19 °C. Such high SSTs during the Ancylus
       Lake phase are in apparent contradiction with what we know about the Holocene in this area, with only minor changes in
       temperature. Therefore, a more likely explanation for the apparently high values of the RI-OH´ index is that they are caused
       by the freshwater conditions and not by high SSTs during the Ancylus Lake phase. This is fully consistent with recent
       observations in Lake Lugano in Switzerland (Sinninghe Damsté et al., 2022). Lipid analysis of the SPM obtained from various
water depths of this stratified and partially anoxic, deep (288 m) lake showed that crenarchaeol and OH-isoGDGTs with 0 to
       2 cyclopentane rings show the same concentration profile, peaking in the deeper (30-100 m) oxic waters well below the
       thermocline but above the oxycline at ca. 100 m. Their origin from thaumarchaea was confirmed by 16S rRNA gene analysis.
       Notably, the RI-OH´ index values for the SPM from the deep oxic waters was 0.87, whereas the OH-GDGTs were produced
       in water with an *in situ* temperature of ca. 6°C, which would translate into temperatures > 20°C (Fietz et al., 2020). This is far
greater than the *in-situ* temperatures at which these OH-isoGDGTs were produced. Sinninghe Damsté et al. (2022) also showed
       that this signal is transported by descending particles to the surface sediments of Lake Lugano, which indicated that further
       research is necessary to determine the usefulness of OH-isoGDGTs as paleotemperature proxies in lake deposits. These
       findings are in good agreement with the apparently far too high values of the RI-OH´ index in the sediments deposited during



the Ancylus Lake phase reported here. It may be that thaumarchaea adjust their membrane lipid composition with respect to

the OH-GDGTs in a different way under freshwater than brackish or marine conditions or that the different salinity resulted in a shift in the Thaumarchaeotal species composition, resulting in a change in the membrane lipid composition. A response to a change in salinity could be consistent with the observation that the values of the RI-OH´ index of the surface sediments of the Bothnian Sea and Bay, where salinities are lower (i.e., 3-6 g kg$^{-1}$) than in other parts of the Baltic Sea, are too high and do not follow the relationship seen for the other surface sediments (Fig. 4d). In the Yoldia Sea phase of the Gotland Basin record

(Fig. 2f), the RI-OH´ index values are lower (0.14-0.58) and fluctuate much more than in the Ancylus Lake phase. This would be consistent with the idea that, despite the somewhat confusing name of the Yoldia Sea phase, the salinity of the water of the basin showed substantial fluctuations between freshwater and brackish (Björck, 1995). The record seems to reveal two periods, i.e., at 470-530 cm and at ca. 410 cm, where the RI-OH´ index values are approaching or are slightly lower, respectively, than those of the brackish period, suggesting that the salinity of the basin was similar as during the brackish period. However, since

the RI-OH´ index is affected by both salinity and temperature, such interpretations should be made cautiously.

In the Arkona Basin record (Fig. 4f) the RI-OH´ index values fluctuate between those of the brackish phase (i.e., 0.25) and substantially higher values (up to 1.03 at the start of the Ancylus Lake phase), generally in line with the observations for the Gotland Basin record. However, the much coarser resolution of sampling makes a more detailed comparison not feasible. For the five datapoints of the Baltic Ice Lake phase, the RI-OH´ index fluctuates between 0.30-0.45 (Fig. 4f). Since this is thought

to have been a freshwater basin that had been filled with water from the melting Scandinavian icecap (Björck, 1995), the lower RI-OH´ index values compared to those of the Ancylus Lake phase would suggest lower water temperatures at that time.

For the portion of the Gotland Basin record when the Baltic Sea was brackish, i.e. from ca. 7.2 cal kyr BP onward, the general trend in SST estimates using the RI-OH´ index is in good agreement with that recorded by the $TEX_{86}^L$ proxy (Fig. 5a) reported earlier (Kabel et al., 2012; Warden et al., 2017). The absolute values of SST and their ranges reconstructed for the brackish

period using the RI-OH´ index (8.6-10.1°C) were smaller than those reconstructed with the $TEX_{86}^L$ (15.3-18.0°C). This is probably because the $TEX_{86}^L$-based calibration reconstructs summer (July-November; Kabel et al., 2012) temperature, whereas RI-OH´ index was calibrated against mean annual SST and absolute values of summer SST and changes therein through time may be larger than for annual mean SST. Both the records obtained by the RI-OH´ index and the $TEX_{86}^L$ capture the major temperature fluctuations during this period, including the Holocene Thermal Maximum (HTM), the Medieval Climate

Anomaly (MCA, from 950-1250 AD) and the Little Ice Age (LIA, from 1400-1700 AD) (Mann et al., 2009), as well as, to some extent, the recent warming trend (starting at the LIA). However, in the upper 11 cm of the record RI-OH´ index values seem to be offset by 0.04 unit (Fig. 5a) for an unknown reason. This is also evident when values of the RI-OH´ index and $TEX_{86}^L$ are cross-correlated (Fig. 5b): they show a linear correlation ($R^2 = 0.76$) when the 0-11 cm section is excluded. Due to the much lower resolution in the brackish section of the Arkona Basin, it is difficult to assess if the RI-OH´ index has captured

the LIA and MCA, although, the recent warming period appeared to be captured (Fig. 3f).

The Ancylus Lake and Yoldia Sea sections of the Gotland Basin record all show much higher RI-OH´ index values but about the same $TEX_{86}^L$ values, except for the datapoints from around 410 cm, which are close to the datapoints from the brackish





phase (Fig. 5b). The six datapoints from the 470-530 cm section of the Yoldia Sea phase with lower RI-OH´ index values (see before) also had the lowest $TEX_{86}^{L}$ values of the whole record (i.e., -0.83 – 0.74), which suggest that the low RI-OH´ index

values in this section are not only the results of increased salinity but also of reduced SST.

## 6 Conclusions

The application of three established OH-GDGT proxies for SST reconstruction, i.e., the OH-GDGT%, RI-OH, and RI-OH´ indices, were tested in a thaumarchaeal enrichment culture, a set of surface sediments and two sedimentary records from the Baltic Sea. The work with the enrichment cultures grown at two temperatures showed that all three proxies show the expected

response with an increase of temperature, but the absolute values obtained for the three OH-GDGT proxies were not always in line with the existing marine core top calibrations. Especially, the OH-GDGT% was much lower than expected, whereas the values of the RI-OH, and RI-OH´ indices was too high at 22°C. The values of the OH-GDGT proxies for the twelve surface sediments studied are generally in line with their variation with SST as shown for global data sets. However, the spread in the SST – OH-GDGT% calibration is so high that reliable paleoSST with this proxy seems implausible. This may, partly, relate

to analytical challenges in the quantification of OH-GDGTs relative to GDGTs. The OH-GDGT SST proxies based only on the distribution of OH-GDGTs generally perform better but the RI-OH index shows no increase with increasing mean annual SST in the surface sediment set of our study. In contrast, the RI-OH´ index shows a linear correlation with SST when the three surface sediments from the Bothnian Sea and Bay are omitted. This appears to be an effect of the substantially reduced salinity in this area although increased lateral sediment influx characteristic for this region in the Baltic Sea may also play a role. In

the brackish phase of the Baltic Sea (the Littorina Sea) the RI-OH´ index overall shows a good correlation with the $TEX_{86}^{L}$, an established temperature proxy in the Baltic Sea, and is able to identify important climatic events, such as the LIA, the MCA, and the HTM. However, during the Ancylus Lake phase values for the RI-OH´ index are far too high, resulting in anomalously high SST estimates. This is probably because thaumarchaea adjust their OH-GDGTs under freshwater conditions in a different way as has recently been shown for thaumarchaea residing in the water column of Lake Lugano. In the Ancylus Lake, the

Baltic Sea, and the Yoldia lake phases of the Baltic basin, the record of the RI-OH´ index, thus, most likely reflects both changes in temperature and salinity. Overall, our study indicates that a reduced salinity increases the values of the RI-OH´ (and RI-OH) indices substantially and this should be considered when applying these for paleotemperature applications in depositional environments where waters may have been fresh for a certain time. Judging from the study of our Baltic surface sediments, this only applies when the surface salinity becomes <6 g kg$^{-1}$. However, since the thaumarchaea producing the OH-

GDGTs reside in deeper waters where salinities are typically higher, it is not possible to determine the salinity at which the use of the RI-OH´ SST proxy becomes complicated.

**Data availability.** Data for this study can be access through the Royal Netherlands Institute for Sea Research (NIOZ) data repository at https://dataverse.nioz.nl/. The dataset can be directly accessed at the following link: https://doi.org/XXXXXX.

**Author contributions. J. S. Sinninghe Damsté**: Conceptualization, Data curation, Formal analysis, Funding acquisition,
Investigation, Methodology, Supervision, Validation, Visualization, Writing - Original Draft, Writing – Review & Editing. **L. Warden**: Conceptualization, Formal analysis, Investigation, Writing – Review & Editing. **C. Berg**: Investigation, Resources, Writing – Review & Editing. **K. Jürgens**: Resources, Supervision, Writing – Review & Editing. **M. Moros**: Conceptualization, Funding acquisition, Investigation, Resources, Supervision, Writing – Review & Editing.

**Competing interests.** The contact author has declared that he nor the co-authors have any competing interests.

**Acknowledgements.** We thank the captains, chief scientists, crews and scientific staff onboard numerous cruises of the R/V *Maria S. Merian*, R/V *Meteor*, R/V *Aranda*, R/V *Prof. Albrecht Penck*, R/V *Poseidon* and R/V *Elisabeth Mann Borgese* for technical assistance with coring. We thank Dr. Martina Sollai for help with extraction and work-up, W. Irene C. Rijpstra for lipid analysis of the archaeal cultures, Jort Ossebaar, Dr. Ellen C. Hopmans, Angelique Metz, and Jessica Rieckenberg for their help with the UHPLC-APCI-MS and data analysis, and Drs. Jerome Kaiser and Susanne Fietz for providing previously 460 published data. Part of the data in this publication has been included in a PhD thesis of one of the authors (Warden, 2017).

**Financial support.** This research was supported by funding from the European Research Council under the European Union's 7th Framework Programme (FP7/2007-2013) / ERC grant agreement n° [226600] to JSSD, and by the Netherlands Earth System Science Center (NESSC) though a gravitation grant (024.002.001) to JSSD from the Dutch Ministry for Education, Culture and Science. The field work received funding from the European Union's 7th Framework Programme (FP/2007–2013) 465 BONUS + under grant agreement 217246.

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



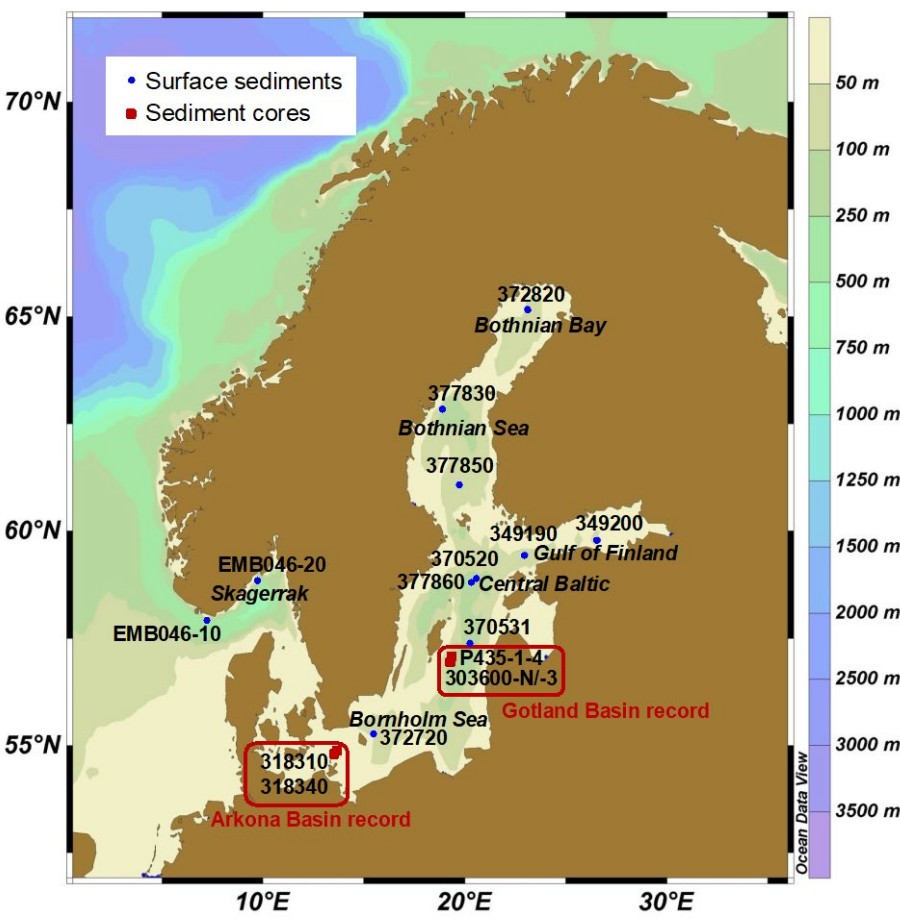

**Figure 1: Map of the Baltic Sea region showing the locations where the surface sediments (indicated by green squares) and sediment cores (indicated by blue circles) were collected. The map was produced by Ocean Data View software (Schlitzer, R., Ocean Data View, http://odv.awi.de, 2016).**





**Figure 2: OH-GDGTs and related information (including a core photograph of the upper section) of the sedimentary record of the Gotland Basin.** Variations with composite depth are shown for (a) TOC content (data from Sollai et al. (2017) and Warden et al. (2017)), concentrations of (b) crenarchaeol (data from Warden et al. (2018)) and (c) summed OH-GDGTs (both in mg g-1 TOC), (d) the OH-GDGT% index, (e) the RI-OH index, and (f) the RI-OH′ index. The average values for the three indices of the nine surface sediments from the Baltic Sea are indicated in (d-f). The various phases of the Baltic Sea are indicated following Sollai et al. (2017). Multicore P435-1-4, core 303600-N, and core 303600-3 sediments are plotted in grey, black, and white circles, respectively. In core 303600-3 sediments, biomarker concentrations were not determined because TOC content data is lacking. In this case, the TOC profile shown in panel (a) was derived from loss on ignition (LOI) data, which were obtained for the whole core at 1 cm resolution (Warden et al., 2017). TOC was calculated by dividing the LOI content by 2.5 for the 0-230 cm interval and by 4.1 for the 230-600 cm interval. These factors were obtained by comparison of the TOC and LOI profiles for core 3030600-N. The higher factor for the part of the core with lower LOI values is due to its higher clay content. Key: MoWP = modern warm period, MCA = the medieval climate anomaly, HTM = the Holocene thermal maximum.







**Figure 3: OH-GDGTs and related information of the sedimentary record of the Arkona Basin.** Variations with composite depth are shown for (a) LOI content (data from Sollai et al. (2017)), concentrations of (b) crenarchaeol (data from Warden et al. (2018)) and (c) summed OH-GDGTs (both in mg g-1 TOC), (d) the OH-GDGT% index, (e) the RI-OH index, and (f) the RI-OH' index. The average values for the three indices of the nine surface sediments from the Baltic Sea are indicated following Warden et al. (2016). Core 31810 and 318340 sediments are plotted in black and white circles, respectively. In the Ice Lake and Yoldia Sea phases, concentrations have not been determined because TOC content data is missing. Key: MCA = the medieval climate anomaly.



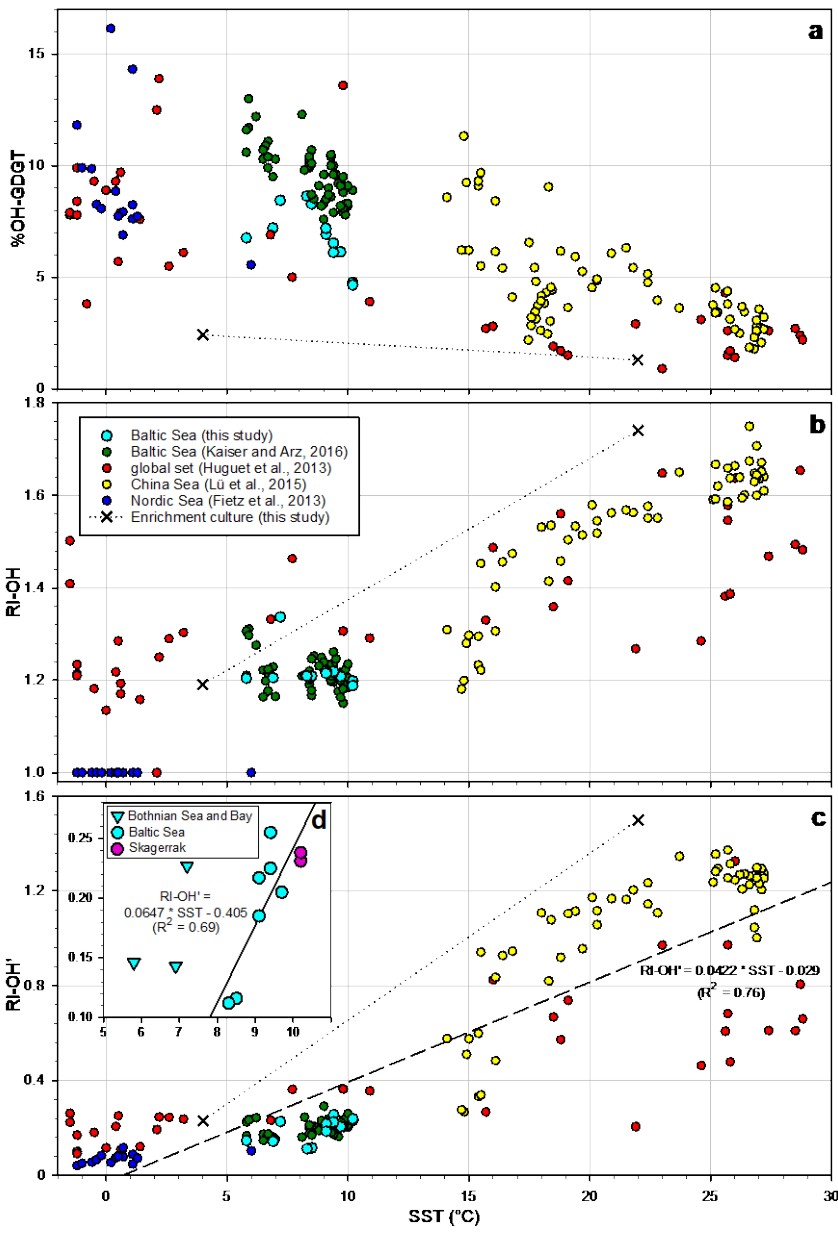

**Figure 4: Scatter plots of the mean annual sea surface temperature (SST; °C) or growth temperature versus (a) the OH-GDGT% index, (b) the RI-OH index, and (c) the RI-OH´ index. The light blue circles represent the surface sediments of the nine surface sediments of the Baltic Sea and two of the Skagerrak (see Table 1) analyzed in this study. Previously reported data of surface sediments from the Baltic Sea and Skagerrak (Kaiser and Arz, 2016), a global data set (Huguet et al., 2013), the China Sea (Liu et al., 2015), and the Nordic Sea (Fietz et al., 2013) are plotted for reference in dark green, red, yellow, and dark blue circles, respectively. Data from the Thaumarchaeotal enrichment culture grown at 4 and 22°C are plotted with crosses and are tied with a stippled line to indicate the trend with temperature. The stippled line in panel (d) is the linear correlation that has recently been proposed as SST calibration for the RI-OH´ (Fietz et al., 2020). The inset (d) in (c) shows an exploded view of the Baltic Sea surface sediment data obtained in this study, revealing a linear correlation between SST and the RI-OH´ index for nine out of twelve datapoints. The three data points obtained from the Bothnian Bay and Sea (indicated with a light blue triangle) were left out of the regression analysis because they are considered as outliers (see text), possibly due to the lower salinity in this area of the Baltic Sea.**



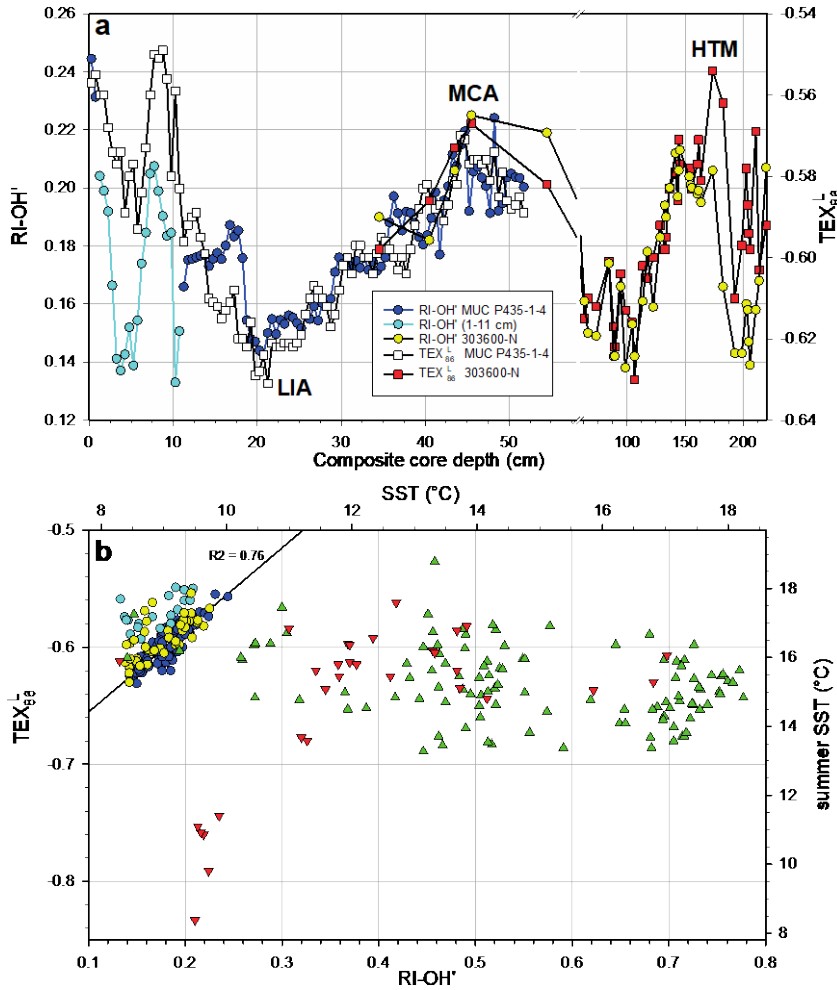

**Figure 5:** The relationship between the RI-OH´ and $TEX_{86}^L$ indices for the Gotland Basin for the studied sedimentary record. (a) RI-OH´ (filled circles) and $TEX_{86}^L$ (filled squares) plotted versus composite core depth for the section covering the Littorina Sea phase. The data of the multicore P435-1-4 are plotted in blue and white symbols, those from piston core 303600-N are plotted with yellow and red symbols. The plot shows the overall good correspondence of variations in the RI-OH´ index and $TEX_{86}^L$ records, suggesting that the RI-OH´ index in the Littorina Sea phase records SST changes. Note that in the upper part of the section (1-11 cm; light blue symbols for RI-OH´) the overall correlation is distorted; the values of the RI-OH´ index seem ca. 0.04 too low. The scale of the composite core depth axis is broken at ca. 60 cm and the scale of the second part of the depth scale is more compressed. For core 303600-N only data points are shown for which both RI-OH´ and $TEX_{86}^L$ values were obtained. Key: MCA = the medieval climate anomaly, LIA = little ice age, HTM = Holocene thermal maximum. (b) RI-OH´ index values plotted vs. $TEX_{86}^L$ values for the complete Gotland basin record. In the left upper corner of the plot the correlation ($R^2 = 0.76$) between the values of the RI-OH´ index and $TEX_{86}^L$ for the P435-1-4 MUC core (dark blue circles) except for the 1-11 cm section (light blue circles) is shown. The samples from multicore 303600-N are plotted with yellow circles (Littorina Sea phase) and green triangles (other phases). Those from the Littorina Sea phase plot in line with the linear correlation line. The samples from the 303600-3 core are plotted with red triangles. The sediments from the other phases (especially in the Ancylus Lake phase, see Fig. 2) show much higher values for the RI-OH´ index, translating into much higher annual mean SSTs (upper scale) using the Baltic Sea calibration (equation 9), whereas those of the $TEX_{86}^L$ are similar or slightly lower and show much less variation in July-November SSTs as estimated with the Baltic Sea calibration (equation 8; Kabel et al., 2012). Note that the Yoldia Sea sediments from around ca. 410 cm plot together with those from the Littorina Sea phase. The other Yoldia Sea sediments with relatively low RI-OH´ index values (470-530 cm; Fig. 2f) show much lower $TEX_{86}^L$ values than all other sediments.

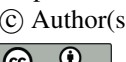

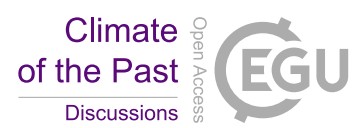

**Table 1** Location, water depth, sea surface temperature (SST) and salinity (SSS) for the stations where surface sediments and sediment cores were obtained and selected (OH)-GDGT data.

| Station | Location | Latitude [N] | Longitude [E] | Water depth [m] | SST[a] [°C] | SSS[b] [g kg⁻¹] | OH-GDGT% | RI-OH | RI-OH' | TEX$_{86}^{L}$ |
|---|---|---|---|---|---|---|---|---|---|---|
| **Surface sediments** | | | | | | | | | | |
| EMB046-10[c] | Skagerrak | 58°49.74 | 07°17.66 | 457 | 10.2 | 34 | 4.8 | 1.198 | 0.231 | -0.653 |
| EMB046-20[c] | Skagerrak | 58°31.59 | 09°29.09 | 532 | 10.2 | 34 | 4.7 | 1.188 | 0.238 | -0.650 |
| 372720[c] | Bornholm Sea | 55°15.67 | 15°28.21 | 96 | 9.7 | 10.5 | 6.1 | 1.208 | 0.205 | -0.591 |
| 370540[d] | Gotland Basin | 57°17.04 | 20°07.26 | 243 | 9.4 | 9.8 | 6.5 | 1.217 | 0.255 | -0.578 |
| 370531[c] | Gotland Basin | 57°23.12 | 20°15.55 | 232 | 9.4 | 9.8 | 6.1 | 1.219 | 0.225 | -0.556 |
| 377860[c] | Central Baltic | 58°48.92 | 20°25.17 | 195 | 9.1 | 8.2 | 6.9 | 1.220 | 0.217 | -0.585 |
| 370520[c] | Central Baltic | 58°53.66 | 20°34.43 | 182 | 9.1 | 8.2 | 7.2 | 1.215 | 0.185 | -0.587 |
| 349190[c] | Gulf of Finland | 59°26.09 | 22°58.39 | 94 | 8.5 | 6.5 | 8.3 | 1.209 | 0.116 | -0.595 |
| 349200[c] | Gulf of Finland | 59°46.97 | 26°35.05 | 82 | 8.3 | 6.5 | 8.6 | 1.209 | 0.112 | -0.579 |
| 377850[d] | Bothnian Sea | 61°04.31 | 19°43.68 | 136 | 7.2 | 5.7 | 8.4 | 1.337 | 0.227 | -0.621 |
| 377830[d] | Bothnian Sea | 62°50.71 | 18°53.34 | 210 | 6.9 | 5.8 | 7.2 | 1.205 | 0.143 | -0.650 |
| 372820[c] | Bothnian Bay | 65°10.70 | 23°05.73 | 122 | 5.8 | 3.4 | 6.8 | 1.204 | 0.146 | -0.664 |
| **Sediment cores** | | | | | | | | | | |
| P435-1-4 MUC | Gotland Basin | 56°57.94 | 19°22.21 | 178 | n.a. | n.a. | 4.4-7.3 | 1.17-1.23 | 0.13-0.25 | -0.63– -0.55 |
| 303600-N | Gotland Basin | 56°55.01 | 19°19.99 | 170 | n.a. | n.a. | 4.1-10.9 | 1.18-1.45 | 0.14-0.78 | -0.69– -0.55 |
| 303600-3 | Gotland Basin | 56°55.01 | 19°19.99 | 170 | n.a. | n.a. | 6.4-9.6 | 1.17-1.43 | 0.13-0.70 | -0.83– -0.56 |
| 318310 | Arkona Basin | 54°50.34 | 13°32.03 | 46 | n.a. | n.a. | 3.4-5.7 | 1.26-1.53 | 0.27-0.92 | -0.60– -0.20 |
| 318340 | Arkona Basin | 54°54.77 | 13°41.44 | 47 | n.a. | n.a. | 1.5-9.5 | 1.23-1.63 | 0.26-1.03 | -0.62– -0.40 |

[a] SST was obtained from CTD measurements and reflects annual mean SST; [b] SSS reflects the average of the years 2000-2006 obtained from the closest HELCOM sampling site (see www.ices.dk); [c] 0-1 cm; [d] 0-0.5 cm; n.a. = not applicable

**Table 2** Fractional abundances of the isoGDGTs (including OH-GDGTs) and GDGT ratios in the Thaumarchaeotal enrichment culture grown at two temperatures.

| T (°C) | GDGT-0 | GDGT-1 | GDGT-2 | GDGT-3 | Crenarchaeol | Cren' | OH-GDGT-0 | OH-GDGT-1 | OH-GDGT-2 | OH-GDGT% | RI-OH | RI-OH' | TEX$_{86}^{L}$ |
|---|---|---|---|---|---|---|---|---|---|---|---|---|---|
| 4 | 66.4 | 2.89 | 0.60 | 0.36 | 27.0 | 0.37 | 1.95 | 0.38 | 0.09 | 2.4 | 1.19 | 0.23 | -0.81 |
| 22 | 23.0 | 22.7 | 18.8 | 2.86 | 30.9 | 0.45 | 0.18 | 0.29 | 0.83 | 1.3 | 1.74 | 1.50 | -0.37 |