# Peer review of "Evaluation of the distributions of hydroxylated isoprenoidal GDGTs in Holocene Baltic Sea sediments for reconstruction of sea surface temperature: The effect of changing salinity"

_Climate of the Past, 2022_

## Author Comment (AC2)

**Rebuttal to the comments of referee #2**

We thank the referee for the positive assessment of our manuscript. Although the referee finds the presented data on the salinity impact on the distribution of OH-GDGTs interesting, he/she raises some concerns. We agree with the referee that the slightly deviating values of the OH-GDGT-derived SST indices in the surface sediments of the northern Baltic alone cannot be considered to be solid evidence for the influence of salinity on the values of RI-OH′ index. The observed deviation may also be caused by an increased lateral sediment influx, as is clearly stated in the abstract and main text of the manuscript. The predominant argument for a potential effect of changing salinity on OH-GDGT-derived SST indices is the huge change that occurs in the values of e.g. the RI-OH′ index during the freshwater (Ancylus) phase (i.e. from 0.18 to ca. 0.75). With the existing global core top calibrations such a high value would translate into an SST of 17°C, which is entirely unrealistic. This is corroborated by the fact that TEX′$_L$ values remain constant or are even slightly lower (see Figure 5). Our observation made at two different locations in the Baltic Sea is entirely consistent with recent observations in a large Swiss freshwater lake (Lake Lugano; Sinninghe Damste et al., 2022). This strongly suggests that when salinity drops below a certain threshold value (which is assumed to be relatively low, e.g. 4 g kg$^{-1}$ or even lower) thaumarchaea adjust the distribution of their (OH)-GDGTs differently as thaumarchaeota growing in brackish or marine waters. We agree with the referee that our dataset does not provide 100% evidence for a causal relationship, but we would like to emphasize that we do not claim this (e.g., "the record of the RI-OH′ index, thus, most likely reflects both changes in temperature and salinity", line 31).

The referee remarks that the salinity "effect" could have been further examined with our thaumarchaeotal enrichment culture. We agree with the referee that additional culture experiments with thaumarchaeotal cultures and enrichments grown at various salinities and temperatures would be a useful topic for futures studies. However, this type of work is extremely laborious, and we consider this beyond the scope of this already "data-rich" study. For our experiment with the enrichment culture just the two temperature "extremes" that the organisms are faced with in the Baltic Sea water column were chosen to see if a significant effect occurs on the (OH)-GDGT distribution. An increase of the number of data point along this temperature gradient, as suggested by the referee, would indeed be of interest and is a promising topic for future studies. Our enrichment culture was grown at a salinity of 12 g kg$^{-1}$. Reduction of the salinity by 70% or more would have very likely led to ceasing of the culture. The growth rate of the enrichment culture was already very low at the low temperatures, characteristic for this environment. So, therefore, it is hardly possible to obtain information on the salinity response within this experimental set-up. Furthermore, we analyzed previously a "natural" enrichment culture in the deeper waters of Lake Lugano growing at 6°C. In this setting, 93-100% of all archaeal 16S rRNA sequences were derived from a single freshwater thaumarchaeon and the (OH)-GDGT distribution was also characterized by an anomalously high values of the RI-OH′ index (Sinninghe Damste et al., 2022). It is hard to come up with a better equivalent for the Ancylus Lake phase of the Baltic Sea than this natural setting, so we feel confident about our suggestion that salinity may have a substantial effect on temperature proxies based on OH-GDGTs.

Another suggestion of the referee was to examine the core-top dataset on the potential effect of salinity on the distribution of OH-GDGTs. Unfortunately, these data are not from areas with a low salinity (<4 g kg$^{-1}$), which would be required to do so.

We have chosen to plot the sediment data versus depth instead of time because the age models are not well constrained for the other phases than the brackish phase. We will add in the figures the estimated age of the boundaries between the various phases (stippled lines in Figures 2 and 3).

In the revised version we will adjust our manuscript according to the other remaining useful minor remarks on the text and figures of the referee. We performed base and not acid hydrolysis because the OH-GDGTs partially loose their hydroxy group upon acid treatment (see Sinninghe Damste et al., 2022 for details). This would possibly alter their distribution.

References

Sinninghe Damsté, J. S., Weber, Y., Zopfi, J., Lehmann, M. F., Niemann, H. (2022) Distributions and sources of isoprenoidal GDGTs in Lake Lugano and other central European (peri-)alpine lakes: Lessons for their use as paleotemperature proxies. Quatern. Sci. Rev. 277, 107352, http://dx.doi.org/10.1016/j.quascirev.2021.107352.

---

## Author Response (AR1)

Dear Prof. Seidenkrantz,

Please find attached the revised version of our manuscript "Evaluation of the distributions of hydroxylated isoprenoidal GDGTs in Holocene Baltic Sea sediments for reconstruction of sea surface temperature: The effect of changing salinity." submitted to Climate of the Past (p-2022-19). We have adjusted the manuscript according to the comments of the three anonymous referees as specified in our detailed rebuttal.

We hope that the revised manuscript is now acceptable for publication in CP.

**Rebuttal**

In this rebuttal we have listed all the comments/remarks of the editor and referees in italic font and our response in normal font.

**Editor**

*As you are aware, your manuscript has been evaluated by three reviewers. I am pleased to see that all reviewers find your study to be interesting and are overall positive in their comments; one reviewer requests only minor revisions, while two asks for major revisions before acceptance. I thus invite you to submit a revised version of your manuscript taking all review comments into accounts.*

*Based on your preliminary replies, I get the impression that there are some of the reviewers' comments that you may prefer not to follow fully. At times, this may be the correct choice, however, it is likely that other readers will raise the same questions as the reviewers and therefore it will be a good idea to shortly explain your choices in the manuscript. So please consider carefully if addition of a few chose words or a short sentence would prevent future confusion.*

It is indeed true that we did not change our manuscript according to ALL the points raised for reasons explained in our replies (see below). If we would do this, it would lead to a substantial decrease in the readability of our manuscript, which cannot be the idea of peer review. In the publicly available response to the comments of the reviews, the interested reader can access this discussion (i.e., that is the advantage of the peer review system used). In this system, the referees have also been offered the possibility to respond to our comments but they have chosen not to do so. Where possible, we have of course, clarified the raised items by adding a few words or sentence(s). This has been specified below.

*As always, when you resubmit, please reply to all reviewers' comments in detail and mark clearly any changes made to the manuscript in the text through highlights or track-changes.*

A manuscript file where the changes made have been marked has been uploaded.

**Referee #1**

*In this study, authors presented results of a Thaumarchaeotal culture enrichment, surface sediments, and two composite sediment cores from the Baltic Sea to assess the use of OH-GDGT-based indices, OH-GDGT%, RI-OH and RI-OH', as sea surface temperature (SST) proxies. Based on their enrichment and surface sediments, those indices (RI-OH' in particular) indeed respond to temperature changes, although it appears that quantitative estimate of temperature remains challenging. However, in their sediment core records, those indices become problematic when the Baltic Sea experienced from the fresh Ancylus Lake to the brackish Littorina Sea phases. The RI-OH' values are exceptionally high during the fresh Ancylus Lake phase. Authors conclude that a reduced salinity could increase the RI-OH' values substantially, and thus this effect should be considered when past salinity changed dramatically.*

*OH-GDGT-based proxies have been proposed to reconstruct past SST, however, confounding factors for these indices have not been fully evaluated, limiting their use as SST proxies. This study presents solid evidence that other factors could also affect those indices. The manuscript is well written and the scientific issue is well articulated and addressed.*

We thank the referee for the positive assessment of our manuscript. We agree that, although OH-GDGT-based proxies are promising, we need to know more about their behavior under varying circumstances before we can fully apply these proxies with confidence. Our data show that at low salinity caution should be exercised.

*I have only one comment. Authors believe that either thaumarchaea adjust their membrane lipid composition differently in freshwater vs. brackish conditions or different salinity could result in a shift in the Thaumarchaeotal species composition and suggest that the salinity effect should be considered when waters became fresh in the past. I'm not sure whether authors could recommend a GDGT-based index, ideally to be retrieved from the same OH-GDGT analysis, to signal this effect, without knowing its salinity history. Based on authors' present and previous study (Warden et al., 2018), it appears that the BIT and MBT'5ME indices also changed correspondingly, associated with the changes in sedimentary environment, which appears to me that the archaea community may have changed substantially in response to the salinity changes. Thus, the BIT index could be used to signal this effect (but I understand that the BIT index does not indicate specific changes in the Thaumarchaeotal species composition)? Just a thought.*

The one question raised by the referee is a valid one and basically falls apart into two sub-questions: 1) Does the large change in distribution derives from membrane adjustment of the same thaumarchaeal species to the salinity change or is it due to a change in the thaumarchaeotal species composition?, and 2) Is there a GDGT-based ratio that can be used to "predict" a salinity change, which could be used to indicate that OH-GDGT-based proxies should be used with caution?

The first question cannot be answered with certainty. It seems likely that the composition of the thaumarchaeotal community will alter with a change in salinity. However, the abundant thaumarchaeotal species residing in the fresh oxic bottom waters just above the chemocline of Lake Lugano (Sinninghe Damste et al., 2022) is phylogenetically very closely related to various *Nitrosopumilus* spp. (98.9% 16S rRNA gene sequence similarity) and Ca. Nitrosarchaeum limnium (96.0% sequence similarity), to which the predominant

thaumarchaeote in the Baltic Sea is closely related. This would suggest a similar response of membrane composition to changing physiological conditions, but a definite answer can only be obtained by testing with cultures. This was already described in the original submission. Unfortunately, we only tested how the membrane composition of the thaumarchaeotal strain isolated from the Baltic Sea responded to temperature but not to salinity. This remains an important challenge for the future.

With respect to the second sub-question: the referee is right that many GDGT ratio's and those of other biomarkers such as glycolipids (Sollai et al., 2017) change at the transition from freshwater to brackish conditions with the change in BIT index as a marked example. Unfortunately, however, none of these can be interpreted with certainty as true indicators of salinity; the changes observed are predominantly caused by large changes in the microbial community composition as a response to the changes in salinity but cannot be used to confidentially indicate that. At present, we believe we should rely on other proxy record (e.g. diatom skeletons, alkenone distributions) to independently assess if salinity changes may have influenced OH-GDGT-based sea surface temperature records.

*Line 25: "are omitted", incomplete phrase*

This has been corrected.

*Line 205: change "as for" to "than"?*

Changed.

**Referee #2**

*This paper presents OH-GDGT distributions in surface sediments and two sediment cores from the Baltic sea and investigated the potential impact of salinity on the OH-GDGT distribution in these samples. The sea surface temperature (SST) was found to be the major factor controlling the RI-OH', and a higher culture temperature results in a higher RI-OH' value in a thaumarchaeal enrichment culture experiment. However, several outliers were found to deviate from the linear correlation between RI-OH' and SST in surface sediments of the Baltic sea, suggesting that a lower water salinity may increase the RI-OH' value. In addition, the RI-OH' index in a sediment core shows a good correlation with the $TEX_{86}{}^{L}$ in the brackish phase of the Baltic sea whereas this index shows abnormally higher values during the freshwater phase. These pieces of evidence collectively suggest salinity should be considered when applying RI-O' to the reconstruction of SST in sediment cores. This paper is generally well written and easy to read. The finding of salinity impact on the distribution of OH-GDGTs is interesting. I have some concerns that need to be addressed.*

We thank the referee for the positive assessment of our manuscript.

*The authors may need more evidence to support the argument that salinity influences the OH-GDGT distribution. Two points in the paper indicate there might be a salinity effect: abnormally high RI-OH' values occur in surface sediments with potentially low water salinity as well as in the freshwater phase of a sediment core in the Baltic sea. These unusually high RI-OH' values were attributed to the low water salinity. However, it remains to be determined whether there is a causal relationship between higher RI-OH' and lower salinity.*

We agree with the referee that the slightly deviating values of the OH-GDGT-derived SST indices in the surface sediments of the northern Baltic **alone** cannot be considered to be solid evidence for the influence of salinity on the values of RI-OH′ index. The observed deviation may also be caused by an increased lateral sediment influx, as is clearly stated in the abstract and main text of the manuscript. Our predominant argument for a potential effect of changing salinity on OH-GDGT-derived SST indices is the huge difference observed between the values of e.g. the RI-OH′ index during the brackish and freshwater (Ancylus) phase (i.e., 0.18 vs. ca. 0.75). With the existing global core top calibrations such a high value would translate into an SST of 17°C, which is entirely unrealistic. This is corroborated by the fact that TEX′$_L$ values remain constant or are even slightly lower (see Figure 5). Our observations which made at two different locations in the Baltic Sea is entirely consistent with recent observations in a large Swiss freshwater lake (Lake Lugano; Sinninghe Damste et al., 2022). This strongly suggests that when salinity drops below a certain threshold value (which is assumed to be relatively low, e.g. 4 g kg$^{-1}$ or even lower) thaumarchaea adjust the distribution of their (OH)-GDGTs in a different way as thaumarchaeota growing in brackish or marine waters. This has been extensively discussed in the original submission. We agree with the referee that our dataset does not provide 100% evidence for a causal relationship, but we would like to emphasize that we do not claim this in our manuscript (e.g., "the record of the RI-OH´ index, thus, most likely reflects both changes in temperature and salinity", line 31 in the original manuscript).

*It would be interesting to see whether there is a close relationship between RI-OH' and salinity in the thaumarchaeal enrichment culture.*

We agree with the referee that additional culture experiments with thaumarchaeotal cultures and enrichments grown at various salinities and temperatures would be a useful topic for futures studies. However, this type of work is extremely laborious, and we consider this beyond the scope of this already "data-rich" study. For our experiment with the enrichment culture just the two temperature "extremes" that the organisms are faced with in the Baltic Sea water column were chosen to see if a significant effect occurs on the (OH)-GDGT distribution. An increase of the number of data point along this temperature gradient, as suggested by the referee, would indeed be of interest and is a promising topic for future studies. Our enrichment culture was grown at a salinity of 12 g kg$^{-1}$. Reduction of the salinity by 70% or more would have very likely led to ceasing of the culture. The growth rate of the enrichment culture was already very low at the low temperatures, characteristic for this environment. So, therefore, it is hardly possible to obtain information on the salinity response within this experimental set-up. Furthermore, we analyzed previously a "natural" enrichment culture in the deeper waters of Lake Lugano growing at 6°C. In this setting, 93-100% of all archaeal 16S rRNA sequences were derived from a single freshwater thaumarchaeon and the (OH)-GDGT distribution was also characterized by an anomalously high values of the RI-OH´ index (Sinninghe Damste et al., 2022). It is hard to come up with a better equivalent for the Ancylus Lake phase of the Baltic Sea than this natural setting, so we feel confident about our suggestion that salinity may have a substantial effect on temperature proxies based on OH-GDGTs.

*Also, the previously published global data set may provide some clues for the impact of salinity on the RI-OH' because those samples were collected from seas with different surface water salinity.*

Unfortunately, these data are not from areas with a low salinity (<4 g kg$^{-1}$), which would be required to do so.

*L20-22 The inconsistency of OH-GDGTs between the thaumarchaeal enrichment culture and core top sediments might be because the dominant thaumarchaeota producing OH-GDGTs in sediments differ from that cultured in this study.*

We cannot fully exclude that but the enrichment culture was obtained from the Landsort Deep of the Baltic Sea (see Berg et al., 2015), which lies in the central Baltic Sea. We mention in the revised manuscript (Lines 287-288) that this is the best possible option.

*L144-145 Why do you use base hydrolysis here? The base hydrolysis can only transform a part of intact polar lipid (IPL) to core lipids (CL) while those polar lipids with glucose headgroup remain intact.*

We performed base and not acid hydrolysis because the OH-GDGTs partially lose their hydroxy group upon acid treatment (see Sinninghe Damste et al., 2022 for details). This would possibly alter their distribution. We have specified this in the experimental section (Lines 149-155). See also the response to a comment of referee #3.

*L146 grounded should be ground*

Corrected.

*L176 Lu should be Lü*

Corrected.

*L186 Why were the thaumarchaeota cultured only at 4 and 22 °C? Culturing this thaumarchaea along a temperature gradient would be better to test the relationship between isoGDGT distribution and temperature.*

See earlier response: this type of work is extremely laborious, and we consider this beyond the scope of this already "data-rich" study.

*L217 and 250 - ~ keep one*

Corrected.

*L294 'grown at' at which temperature?*

Both growth temperatures are mentioned in this sentence.

*L350 than that of .....*

"of OH-GDGTs" was added to clarify the sentence.

*Figure 1 I am confused about the legend and the illustration of the figure. I did not see green squares but only see red squares, which represent sediment cores in the legend. Also, a scale bar and the north direction should be presented.*

Green squares have been corrected to red squares in the legend. The map provides longitude and latitude and a depth scale for water depth (now indicated in the caption), so we consider it overdone to provide 'a scale bar and the north direction' as requested by the referee.

*Figure 2 and 3  I do not know what the grey lines indicate. If they are not used to highlight anything important, I suggest the authors remove these grey lines to make a clearer figure.*

The grey lines were just meant to help the reader but indeed make the figure busy and have been removed following the suggestion of the referee.

*Figure 4   Liu et al. 2015  should be Lü et al.,2015*

Corrected.

*I have noticed the sediment cores used in this study have been dated and described in Warden et al. (2016). However, core depths, instead of the ages, were presented in the figures of this study. That is not helpful for interpretation of the SST change if the time frame is not well constrained.*

We have chosen to plot the sediment data versus depth instead of time because the age models are not well constrained for the other phases than the brackish phase. We have added in the figures the estimated age of the boundaries between the various phases (stippled lines in Figures 2 and 3).

**Referee #3**

*Sinninghe Damsté et al. describe the distribution of OH-GDGT lipids in Baltic Sea surface sediments, sediment cores and a thaumarchaeal culture grown at different temperatures. The manuscript is thorough and well-written and many significant, open questions on OH-GDGT proxies are addressed by the authors.*

We thank the referee for this positive assessment of our manuscript and the useful comments.

*However, there are some issues with the methodology that could complicate interpretation of the results. First, the authors used base hydrolysis to degrade intact polar OH-GDGTs into core lipids, presumably to facilitate analysis. While this method preserves the hydroxyl groups compared to acid hydrolysis, it does not quantitatively remove the headgroups of thaumarchaeal IPLs (Schouten et al., 2008, AEM) and may selectively degrade IPLs according to their headgroup composition. This will lead to biases in the resulting OH-GDGT quantifications and indices because it is known that OH-GDGTs mostly occur in the form of glycosidic IPLs rather than phosphatidic IPLs in Thaumarchaeota (Schouten et al., 2008, AEM; Pitcher et al. 2011, AEM; Elling et al., 2014, GCA), which are not significantly degraded to core lipids by base hydrolysis. Further, the proportion of core structures is different for different IPL types and markedly different between core OH-GDGTs and OH-GDGTs (predominance of OH-GDGT-0) and IPL GDGTs (predominance of OH-GDGT-2) in thaumarchaeal cultures including the strain most closely related to the organism used in this study, N. limnia (Elling et al., 2014 GCA; Elling et al. 2017 GCA; Pitcher et al., 2011 AEM). It is unclear why the lipids were not analyzed as IPLs, which would have circumvented these issues. If aliquots of the lipid extracts are still available, I would recommend re-analyzing the samples using a method appropriate for detection of IPLs.*

The referee is right that the distribution of OH-GDGTs in different GDGT classes (core lipids, phospholipids, glycolipids) may be different. In our analytical approach, we applied base hydrolysis which results in the hydrolysis of phospholipids but leaves the glycolipids intact. This approach was chosen for two reasons. (i) Upon base hydrolysis the OH-GDGTs remain intact, whereas acid hydrolysis (which would also hydrolyze glycolipids) would result in (partial) removal of the hydroxy group of the OH-GDGT, rendering the assessment of their distribution impossible. (ii) This approach results in the "summed" distribution of core and phospholipids GDGTs, which likely comes close to the distribution observed in surface sediments since phospholipids are much more prone to hydrolysis in the environment than glycolipids (Schouten et al., 2010). After all, for this paper, we are interested to compare our culture results with those of the distributions of the core OH-GDGTs in sediments reported in earlier studies which used OH-GDGTs as a paleotemperature proxy. The referee advises to analyze the IPLs as such. Unfortunately, this is not possible as these experiments were performed >10 years ago and no cell material is available anymore. Furthermore, the quantitative extraction of distributions of OH-GDGTs from such data will be extremely complicated since response factors for GDGT IPLs with different head groups and containing OH-GDGTs as cores are unknown and likely to vary substantially, which would make such an exercise difficult and the results would be hard to compare with OH-GDGT distributions in sediments, the aim of these experiments. We have added a few lines to explain our experimental approach (lines 149-154 and 296-299) in the revised manuscript.

*Second, the authors quantified the fragmentation products of OH-GDGTs at m/z 1300, 1298, 1296 and did not include the 1318,1316, 1314 ions, which could explain some of the variation compared to previous studies and during replicate analysis due to variations in ionization efficiencies due to. This approach was used by some previous studies (e.g., Lü et al., 2015, OG), i.e., in contradiction to what is stated in lines 159-161 and 321-322.*

With the APCI method we applied, the OH-GDGTs hardly (<1 %) generate MH+ ions (m/z 1318,1316, 1314) and the MH+-18 ions predominate, so it does not really make sense to include them in the integration. This is now mentioned in the manuscript (Lines 168-171). The referee is right that in the studies of the Bremen group the MH+ ions were taken into account in the integration, and we have this accordingly. In other studies, also only the MH+-18 ions were taken into consideration. In our opinion, these slightly different methods in quantification, will not make a significant difference. In fact, these different methods of quantification have not prevented the combination of data acquired with both quantification methods in earlier important studies for the development of OH-GDGT-based palaeoproxies (see Lü et al., 2015; Fietz et al., 2020).

*Third, the authors do not provide sufficient information on the growth of the cultures such as detailed growth conditions (e.g., volume of medium, volume and type of container and closure, substrate concentration, medium composition) or growth data (e.g. growth curves) to allow assessment of the influence culture conditions on lipid profiles. For instance, it is probable that growth at 4 °C was much slower than at 22 °C. It is known that growth limitation and growth rate influence GDGT cyclization (Qin et al., 2015, PNAS; Hurley et al., 2016, PNAS) and thus it is probable that these factors influenced the presented OH-GDGT data in addition to the influence of temperature. These issues need to be discussed in the manuscript and their potential impact on the results from the culture sample needs to be quantified or acknowledged.*

Details on the experimental conditions to maintain and grow the enrichment culture were highly similar to those reported in Berg et al. (2015). We refer more explicitly to this in the

revised text (lines 119-121) but in our view, there is no need to repeat all experimental details. At both growth temperatures the enrichment cultures were grown in batch cultures until reaching the stationary phase as monitored by FISH/DAPI staining and microscopical observations. In our exploratory experiments, the influence of growth rate has not been studied. Considering the slow growth of the enrichment culture (>40 days until reaching stationary phase at 22°C; Berg et al. 2015), and an even lower growth rate at 4°C, it would constitute an enormous effort to precisely assess growth rate and its impact on GDGT data. Thus, this was entirely beyond the scope of the present study and would reflect a full research project on its own. As requested by the referee, however, we will acknowledge in the revised paper that differences in growth rate may affect the distribution of OH-GDGTs in addition to temperature (Lines 122-123).

*Line comments:*

*Line 18: Missing word? OH-GDGT [indices] for SST reconstruction?*

Corrected to "OH-GDGT proxies".

*Line 145: Please add the original citation of the extraction method, not just a derivative.*

We agree with the referee that preferably the original reference should be quoted. However, "Bligh Dyer extraction" is now such an established term that a quote to their 1959 paper (with over 60,000 citations) is in our opinion no longer required. The appreciation of their work comes with its name after the two authors of this paper (Bligh and Dyer, 1959). We refer to Pitcher et al. (2011) because we performed the extractions exactly according to the protocol described in that paper.

*Line 225-227: This is confusing. If the concentrations of OH-GDGTs could not be determined, then how was %OH-GDGT calculated?*

Absolute concentrations of both iso-GDGTs and OH-GDGTs could not be determined because an internal standard was not used for the sediments from this section. However, using the integrated peak areas of iso-GDGTs and OH-GDGTs, %OH-GDGT could still be calculated.

*Figure 1: There are several issues with the legend and caption. The caption states that surface sediments are green squares, but the map and legend show blue circles. The caption further states that sediment cores are blue circles but they are red squares in the map and legend. The red text on brown background in the map are hard to read. Consider coloring land areas grey.*

We have adjusted the Figure according to most of these comments. We kept the land area brown but changed the red text to white to increase the contrast.

*Figures 2-5 are very low resolution and hard to read. Adjust?*

We are confused as the resolution of our original figures is high. Indeed, when the pdf file produced by Copernicus is examined, the resolution is not as good as the original figures but we do not consider this as "very low". We feel that these problems will be solved when the figures appear at the right resolution in the final pdf file. The referee is right that the figures do plot a lot of data and therefore are "busy". As suggested by referee#2 we have removed the

grey lines in Figs. 2 and 3. Otherwise, each panel provides important information that cannot be missed.

*Figure 4: The colors for the "Baltic Sea (Kaiser and Arz, 2016)" and "global set" should be changed. They are indistinguishable for red/green colorblind people.*

This is now fixed by changing the symbols and colors of the symbols using the IBM Design Library 5 color palette for colorblind people. In this way the figure should also be readable for everybody. The figure legend has been adjusted accordingly.

*Figure 5b: This panel needs a legend displayed on the figure. The colors of the triangles are indistinguishable for red/green colorblind people.*

An additional (partially overlapping) legend has been added to Fig. 5b. Again, colors have been adjusted using the IBM Design Library 5 color palette for colorblind people. The figure legend has been adjusted accordingly.

**References**

Berg, C., Listmann, L., Vandieken, V., Vogts, A., and Jürgens, K. (2015). Chemoautotrophic growth of ammonia-oxidizing Thaumarchaeota enriched from a pelagic redox gradient in the Baltic Sea. Frontiers in Microbiology, 5, 786.

Bligh, E. G., and Dyer, W. J. (1959). A rapid method of total lipid extraction and purification. Canadian journal of biochemistry and physiology. 37, 911-917.

Fietz, S., Ho, S.L. and Huguet, C. (2020) Archaeal membrane lipid-based paleothermometry for applications in polar oceans. Oceanography 33, 104-114.

Lü, X., Liu, X. L., Elling, F. J., Yang, H., Xie, S., Song, J., Li X., Yuan H., Li, N. and Hinrichs, K. U. (2015) Hydroxylated isoprenoid GDGTs in Chinese coastal seas and their potential as a paleotemperature proxy for mid-to-low latitude marginal seas. Org. Geochem. 89, 31-43

Pitcher, A., Hopmans, E. C., Mosier, A. C., Park, S. J., Rhee, S. K., Francis, C. A., and Sinninghe Damsté, J. S. (2011). Core and intact polar glycerol dibiphytanyl glycerol tetraether lipids of ammonia-oxidizing archaea enriched from marine and estuarine sediments. Applied and environmental microbiology. 77, 3468-3477.

Schouten, S., Middelburg, J. J., Hopmans, E. C., and Sinninghe Damsté, J. S. (2010). Fossilization and degradation of intact polar lipids in deep subsurface sediments: A theoretical approach. Geochimica et Cosmochimica Acta 74, 3806-3814.

Sinninghe Damsté, J. S., Weber, Y., Zopfi, J., Lehmann, M. F. and Niemann, H.: Distributions and sources of isoprenoidal GDGTs in Lake Lugano and other central European (peri-)alpine lakes: Lessons for their use as paleotemperature proxies. Quatern. Sci. Rev. 277, 107352, http://dx.doi.org/10.1016/j.quascirev.2021.107352, 2022.

Sollai, M., Hopmans, E. C., Bale, N. J., Mets, A., Warden, L., Moros, M. and Sinninghe Damsté, J. S.: The Holocene sedimentary record of cyanobacterial glycolipids in the Baltic Sea: an evaluation of their application as tracers of past nitrogen fixation. Biogeosciences 14, 5789-5804, http://dx.doi.org/10.5194/bg-14-5789-2017, 2017.

---

## Author Response (AR2)

Dear Prof. Seidenkrantz,

Please find attached the 2nd revised version of our manuscript "Evaluation of the distributions of hydroxylated isoprenoidal GDGTs in Holocene Baltic Sea sediments for reconstruction of sea surface temperature: The effect of changing salinity." submitted to Climate of the Past (p- 2022-19). We have adjusted the manuscript according to remaining comment of Referee #3 as specified in our detailed rebuttal.
We hope that the revised manuscript is now acceptable for publication in CP.

**Rebuttal**
In this rebuttal we have listed all the comments/remarks of the referee in italic font and our response in normal font.

**Referee #3**

*In the new version the authors sufficiently addressed most criticisms.*

We thank the referee for this positive assessment.

*However, one major point still stands. The authors maintain that OH-GDGTs from base hydrolysis of culture biomass are representative of the sedimentary OH-GDGT pool. I find this to be very unlikely due to their biological sources in the environment and find the argumentation by the authors not convincing. This is for two reasons, 1) that most (i.e., close to 90%) of lipids in Thaumarchaeota are glycolipids and not phospholipids 2) that the composition of the lipid pool in the water column and surface sediments is similarly dominated by glycolipids. The authors state that their results are representatives because phospholipids are degraded faster than glycolipids. Even though that is true, this differential degradation cannot account for sedimentary signals. If that were the case, there should be more glyco-GDGTs in sediments than core GDGTs (if the latter are predominantly derived from phospholipids) because phospholipids are so rare in planktonic and benthic archaea. This is clearly not the case given the many studies on IPL-GDGTs in sediments and water columns that found the opposite to be true, including many works by the lead author. Therefore, base hydrolysis cannot be representative of the average lipid composition (or OH-GDGT composition) of the culture or of sediments. I recommend the authors include a more extensive, critical and honest assessment of the caveats of their approach. As a minor point, the citation of the theoretical study of Schouten et al. (2010) could be accompanied by additional references to environmental studies such as Lengger et al. (2012, GCA).*

We thank the referee for sharing once more his/her thoughts on this topic. However, an important topic that the referee fails to see is the fact that all calibration studies based on OH-GDGTs have only taken the core OH-GDGTs in the surface sediments (typically the first 1 cm) into account. So, for these calibrations it is not really relevant in what form the OH-GDGTs reside in the water column or surface sediment (glycolipid or phospholipid fraction) since only the OH-GDGTs in the free (core) form have been analyzed and used for SST calibrations. Of course, this is of concern since during subsequent diagenesis glycolipids may release core OH-GDGTs, which, if they would have a different distribution, may alter the distribution of the core OH-GDGTs.

The comments of Referee #3 on the earlier version of this manuscript made us aware that it would be nice to present also data on the distribution of the IPL of the two enrichment cultures grown at different temperature. After a detailed search of old LC-MS files, which were run in 2011, we came across the analyses of the IPLs of these two cultures, the results of which we have now included in Table 2. Although quantitative data (i.e., distributional data) should be cautiously interpreted because of the strongly varying response factors, these data confirm the trends we see in the acid (see below) hydrolysates and we have included this data in the manuscript (with a clear description of the caveats). We feel that this solves the last issue raised by referee #3 because it basically shows that the OH-GDGT indices have similar values in both fractions.

This search for the original data also revealed that there was an error in the experimental description of the work-up procedure of the two cultures in the PhD thesis of Lisa Warden (which formed the basis for this manuscript); we checked the original notebook of the technician who performed this experimental work and it became clear that the Bligh Dyer fractions were *acid* and not *base* hydrolysed! (In line with all our other research on lipids in archaea from that time). The acid hydrolysis method results in the release of all polar groups from IPL-GDGTs, although partial dehydration of OH-GDGTs occurs. This is now clearly described in the revised manuscript. This, together with the new results on the distribution of the intact IPL-GDGTs, now hopefully will satisfy Referee #3. Because of this, discussion of the different forms of OH-GDGTs (glycol- vs. phospholipids) is not relevant anymore and we have omitted these descriptions.

We thank this referee for her/his persistence on this topic because it prevented us from making a significant error.